


# Optical and geometrical aerosol particle properties over the United Arab Emirates

Maria Filioglou[1], Elina Giannakaki[1,2], John Backman[3], Jutta Kesti[3], Anne Hirsikko[3], Ronny Engelmann[4], Ewan O'Connor[3], Jari T.T Leskinen[5], Xiaoxia Shang[1], Hannele Korhonen[3], Heikki Lihavainen[3,6], Sami Romakkaniemi[1] and Mika Komppula[1]

[1] Finnish Meteorological Institute, Kuopio, FI70211, Finland
[2] Environmental Physics and Meteorology, Faculty of Physics, National and Kapodistrian University of Athens, Athens, GR 15784, Greece
[3] Finnish Meteorological Institute, Helsinki, FI00560, Finland
[4] Leibniz Institute for Tropospheric Research (TROPOS), Leipzig, DE04318, Germany
[5] University of Eastern Finland, Kuopio, FI70211, Finland
[6] Svalbard Integrated Arctic Earth Observing System, Longyearbyen, N-9170, Norway

*Correspondence to*: Maria Filioglou (maria.filioglou@fmi.fi)

**Abstract.** One-year of ground-based night-time Raman lidar observations have been analysed under the Optimization of Aerosol Seeding In rain enhancement Strategies (OASIS) project, in order to characterize the aerosol particle properties over a rural site in the United Arab Emirates. In total, 1130 aerosol particle layers were detected during the one-year measurement campaign which took place between March 2018 and February 2019. Several subsequent aerosol layers could be observed simultaneously in the atmosphere up to 11 km. The observations indicate that the measurement site is a receptor of frequent dust events but predominantly the dust is mixed with aerosols of anthropogenic and/or marine origin. The mean aerosol optical depth over the measurement site ranged at $0.37 \pm 0.12$ and $0.21 \pm 0.11$ for the 355 and 532 nm, respectively. Moreover, a mean lidar ratio of $43 \pm 11$ sr at a wavelength of 355 nm and $39 \pm 10$ sr at 532 nm was found. The average linear particle depolarization ratio measured over the course of the campaign was $15 \pm 6$ % and $19 \pm 7$ % at 355 nm and 532 nm wavelengths, respectively. Since the region is both a source and a receptor of mineral dust, we have also explored the properties of Arabian mineral dust of the greater area of United Arab of Emirates and the Arabian Peninsula. The observed Arabian dust particle properties were $45 \pm 5$ ($42 \pm 5$) sr at 355 (532) nm for the lidar ratio, $25 \pm 2$ % ($31 \pm 2$ %) for the linear particle depolarization ratio at 355 (532) nm, and $0.3 \pm 0.2$ ($0.2 \pm 0.2$) for the extinction-related Ångström exponent (backscatter-related Ångström exponent) between 355 and 532 nm. This study is the first to report comprehensive optical properties of the Arabian dust particles based on long-term observations, using at the fullest the capabilities of a multi-wavelength Raman lidar instrument. The results suggest that the mineral dust properties over the Middle East and western Asia, including the observation site, are comparable to those of African mineral dust with regard to the particle depolarization ratios but not for lidar ratios. The smaller lidar ratio values in this study compared to the reference studies are attributed to the difference in the geochemical characteristics of the soil originating in the study region compared to Northern Africa.



## 1 Introduction

The Earth's energy budget involves the exchange of energy between three levels: its surface, the top of the atmosphere and the
atmosphere in between (Hansen et al., 2005). In this system, aerosol particles are an important, yet underdetermined,
component introducing uncertainties in weather and climatic predictions (Boucher et al., 2013; Stevens and Feingold, 2009).
Additionally, aerosol particles are tied to health (Davidson et al., 2005), biological processes (Kanakidou et al., 2018; Moore
and Braucher, 2008) and aviation safety (Guffanti et al., 2010; Lechner et al., 2017). Mineral dust is one of the most mass
abundant types of primary aerosol particles emitted into the atmosphere (Kok et al., 2017). It accounts for almost 30 to 50 %
of the total global aerosol mass burden and its physicochemical properties such as size distribution, composition, and shape
vary substantially. Recent studies have shown that fine mode dust have a cooling effect on the global climate whereas coarse
dust (particle diameter larger than 5 μm) likely has a warming impact (Kok et al., 2017; Miller et al., 2006). Mineral dust
particles are characterized as nonspherical with irregular shapes and substantial surface heterogeneity (Wagner et al., 2012;
Wiegner et al., 2009). Their optical properties, such as the linear particle depolarization ratio, is also subject to their chemical
composition. Therefore, dust particles originating from different regions exhibit different scattering properties due to their
different microphysical and chemical composition (Järvinen et al., 2016; Müller et al., 2007; Nisantzi et al., 2015; Shin et al.,
2018).

Mineral dust and other aerosol particle types can affect clouds and their microphysical properties and precipitation patterns by
acting as cloud condensation nuclei (CCN) and ice nuclei (IN)  (DeMott et al., 2003; Karydis et al., 2011). To this end,
numerous studies have identified the complex interplay of aerosols and clouds (Morrison et al., 2005; Rosenfeld, 2018). Li et
al. (2017) report that dust-mixed ice clouds have warmer cloud top temperatures (CTTs) suggesting their efficiency to act as
IN. Most recent studies, however, stress the complexity of dust to IN mechanism and its relative effectiveness in different
geographic locations (Ansmann et al., 2009; Coopman et al., 2018; Filioglou et al., 2019; Zamora et al., 2017). The complexity
of dust particles also becomes evident when comparing observations from remote sensing instruments with modelled dust
properties (Binietoglou et al., 2015). Modelling the dust shape and further calculate its optical properties such as dust optical
depth, rely among others on approximations on the sphericity of the dust particles and assumptions on the contribution of
non-dust particles together with vertical dust height information (Dubovik et al., 2002; Hoshyaripour et al., 2019). Accurate
knowledge of the dust optical properties and their spatial distribution in regional and vertical scale is, therefore, a step towards
a more realistic understanding of the climatic forcing impact of this component.

The Middle East and the Arabian Peninsula are one of the major source areas of mineral dust particles, together with northern
Africa. Although this region is key to improving the understanding of the climate impact of mineral dust, very few
measurement campaigns have been conducted and continuous aerosol observations are scarce in the area. In addition of being
one of the world's largest sources of mineral dust, the Arabian Peninsula is also a large emitter of anthropogenic pollution
(Rushdi et al., 2017).  The United Arab Emirates (UAE) is a crossroad for air masses originating from western and central



Asia, or from North Africa (Wehbe et al., 2019). Local emission of mineral dust is also abundant in this area. Regarding anthropogenic pollution, ever growing energy demand have increased $CO_2$ emissions and other pollutants of anthropogenic origin over the past decade (Betancourt-Torcat and Almansoori, 2015; Ukhov et al., 2018) with adverse health effects (Li et al., 2010). These varying aerosol sources make the UAE an interesting area to study aerosol particles, and in particular, dust properties. A few studies indicate that long-range transported dust from the Middle East exhibit different optical properties to

that from Saharan origin (Hofer et al., 2017; Mamouri et al., 2013; Müller et al., 2007; Nisantzi et al., 2015).

To shed further light into atmospheric aerosol properties in the UAE region, a one-year field campaign was conducted from March 2018 to February 2019. The measurement campaign focused on the characterization of the geometrical and optical properties of atmospheric aerosol particles and their interaction with the regional/local meteorology and cloud precipitation patterns under different atmospheric conditions. With less than 100 mm of annual rainfall (Wehbe et al., 2017), precipitation

enhancement techniques such as cloud seeding (French et al., 2018; Vonnegut and Chessin, 1971), have been implemented within UAE's strategy to tackling water shortages in the region. This approach requires accurate understanding of local/regional meteorology, detailed characterization of the background aerosol particles and their efficiency to act as CCN/IN, and the complex interplay between aerosol-cloud-meteorology. Therefore, the Optimization of Aerosol Seeding In rain enhancement Strategies (OASIS) project aimed towards a more robust knowledge of the efficiency of the aerosol particles to

act as CCN/IN in a challenging environment. A multi-instrument approach was used for this purpose including both in-situ and remote sensing sensors along with model simulations. In this paper, we will focus on the characterization of the aerosol properties over the measurement site. Observations of a multi-wavelength Raman lidar with water vapor capability were used along with air mass back-trajectories calculated from the Hybrid Single Particle Lagrangian Integrated Trajectory (HySPLIT) model (Stein et al., 2015) in order to identify and classify the aerosol layers during the campaign period. Moreover, the optical

properties of the Arabian have been characterized.

## 2 Methodology

### 2.1 The measurement site

Between March 2018 and February 2019 the OASIS campaign was established at a palm plantation located 10 km south-west of Al Dhaid city, in the emirate of Sharjah in the UAE (25°14'7.8" N, 55°58'39.97" E, 165 m a.s.l). This rural site is located

at a desert area about 70 km north-east from Dubai and the Arabian Gulf, where oil extraction and shipping activities are situated. To the east, the site faces a mountainous area whose altitude ranges from 1 to 2.1 km, and the sea (Gulf of Oman and Arabian Sea, respectively) (Fig. 1a). In principle, the measurement site receives dust from three different sources. To the North, including Iraq and the surrounding countries, is a region with several sources of dust and the sediment surface may contain sand deposits with particle sizes which are easily lofted by winds. In fact, it is the largest source of Aeolian dust in the Arabian

Gulf. North-east in Iran and Pakistan are regions responsible for dust and sandstorms in Asia. Lastly, Saudi Arabia and the





Arabian Peninsula provide the third major dust source with multiple terrain types. Towards the west side from the measurement location, mountains up to 2.5 km form a natural barrier between this region and the Red Sea. The region itself can be considered as a fourth dust source where dust can be emitted locally due to thermal lows, unstable conditions, or human activities. Anthropogenic pollution is also present in the greater area where oil and gas extraction activities add up to the man-made

aerosol particulate burden from the cities. The Aerosol Optical Depth at 500 nm in the region varies between 0.4 and 0.5 (Eck et al., 2008) where the contribution of mineral dust particles can be 60 - 70 % even in urban areas (Roshan et al., 2019). Figure 1b shows the air mass backward trajectory cluster analysis, computed with HySPLIT (Stein et al., 2015, see Section 2.3), and their frequency over the course of the campaign period. The aforementioned aerosol sources can be viewed at the location of the backward trajectory paths.

## 2.2 The multi-wavelength Raman lidar FMI - Polly$^{XT}$

The FMI-Polly$^{XT}$ lidar is a fully automated instrument capable of 24/7 operation (Engelmann et al., 2016). It is equipped with three elastic backscatter channels at 355 nm, 532 nm and 1064 nm, two rotational-vibrational Raman channels at 387 nm and 607 nm, two linear depolarization channels at 355 nm and 532 nm and one water vapor detection channel at 407 nm. In addition to the far field capabilities, the system includes two near field elastic backscatter channels at 355 nm and 532 nm and two near

field rotational-vibrational Raman channels at 387 nm and 607 nm. Due to the near field capability, full overlap is attained at around 120 m. Data are acquired with a vertical resolution of 7.5 m in temporal steps of 30 s.

The lidar has been employed under various campaigns and locations over the course of years. Among others, two long-term aerosol experimental campaigns at Gual Pahari, India (Komppula et al., 2012) and Elandsfontein, South Africa (Giannakaki et al., 2015, 2016; Korhonen et al., 2014) and at the permanent measurement site in Vehmasmäki, Finland (Bohlmann et al.,

2019; Filioglou et al., 2017) have been conducted. The system is also part of the Finnish lidar network (Hirsikko et al., 2014), the European Aerosol Research Lidar Network (EARLINET) (Bösenberg et al., 2003; Pappalardo et al., 2014) and PollyNET (Baars et al., 2016) which is an independent Raman and polarization lidar network where measurements from all member-stations are visualized through "quick looks", publicly available on the web page of PollyNET (http://polly.tropos.de).

### 2.3 Processing of lidar observations

For the analysis presented, two aerosol profiles were retrieved per day using the Raman method (Ansmann et al., 1990, 1992; Whiteman, 2003). The temporal averaging of each profile corresponded to two hours. In total, 380 profiles were retrieved at fixed times each day (01 and 20 UTC) in order to derive all possible optical properties minimizing the assumptions in the retrievals. The 2-hour average profiles were further analysed by detecting intensive aerosol layers and isolating them from air segments containing very low aerosol particle burden. For the automatic detection of the aerosol particle layers we used the

second derivative of the backscatter profiles. In total 1130 high quality aerosol particle layers were detected during the campaign period. We considered as high quality aerosol layers the ones which were not affected by clouds and exhibited lidar



ratios between 5 and 150 sr, linear particle depolarization ratio between 0 and 40 % and Ångström exponents between -1.1 and 3. The geometrical boundaries of the aerosol particle layers were retrieved from a less vertically smoothed lidar profile (less than 400 m) as opposed to the optical properties which were retrieved by applying higher smoothing (case depended). By

applying less smoothing to the signals, we were able to appoint correct geometrical depth and boundaries of these layers while the higher smoothing assigned meaningful optical properties. Mean values of all the available optical properties, i.e backscatter ($\beta$) and extinction ($\alpha$) coefficients, lidar ratios (LR), Ångström exponents (AE for the extinction-related and BAE for the backscatter-related Ångströms), color ratios (CR), linear particle depolarization ratios ($\delta_P$) and aerosol optical depths (AOD), were then calculated for each of the layers, along with their geometrical properties (depth and centre mass). A 5-day backward

trajectory analysis at the centre mass of each of the layers was also computed using HySPLIT in order to assess the origin of the detected aerosol particle layers. The timestamp used for the trajectories was the centred 2-hour lidar retrieval.

## 2.4 Microanalysis of the collected dust particles

To aid the findings in Section 3.3 where we retrieve the optical properties of the Arabian dust particles we have collected two

dust samples. The samples were dry collected from two different locations around the measurement site where different macrophysical properties, e.g. color, were evident. Two particle distributions were studied to reveal physicochemical properties of gathered particles, i.e., size, morphology and composition. In order to analyse the chemical composition of the particles, energy dispersive X-ray spectroscopy (EDX, Thermo Pathfinder 1.4, Thermo Fisher Scientific, Madison, WI, USA) was used in synergy with a scanning electron microscope (SEM, *Zeiss SigmaHD/VP, Carl Zeiss NTS, Cambridge, UK*) which

was used to observe the morphology of the dust particles. For this, dust samples were attached on a standard 12 mm aluminium stub for SEM specimens using a piece of double sided carbon adhesive tape. The SEM imaging was executed without any sputter coating in low vacuum (Zeiss Variable Pressure mode), Nitrogen atmosphere at 30 Pa pressure using 15 kV acceleration voltage and variable pressure secondary electron (VPSE) detector and a working distance of 15 mm. The elemental composition for individual particles was obtained using EDX mapping. The chemical analysis and two SEM images can be

found in Appendix A.

## 3 Results

### 3.1 Geometrical properties and aerosol optical depths of aerosol particle layers

Altogether 1130 night-time aerosol particle layers have been analysed throughout the campaign period in order to characterize the background aerosol properties over the measurement site. The time series of the geometrical extent of the retrieved aerosol

particle layers showed up to 7 simultaneous layers (Fig. 2). Indeed, as observed in the dataset, frequent multiple aerosol particle layer structures were present most of the time, with single-layers mostly occurring during December and January. In fact, only 10% of the cases had a single aerosol layer present, with two (30 %), three (29 %) or even more simultaneous layers (31 %).





The multiple aerosol particle layers result from gravitational waves generated by the sea breeze passing over the mountains, stratifying the atmosphere over the measurement site. The gaps in the dataset during May to August and between September and November were due to instrumental malfunction, mainly failure of the cooling unit of the system while performing under such demanding conditions where maximum ambient temperatures up to 51 ℃ were measured.

Geometrical features of the aerosol particle layers are further characterized by frequency distributions (Fig. 3). Up to 61 % of the layers identified were located below 2.5 km in altitude, with few layers reaching as high as 11 km. The geometrical depth of the layers varied from a few hundred metres to several kilometres throughout the period. Most commonly (58 % of the cases) the geometrical depth varied between 0.4 and 0.8 m.

In order to define the geometrical boundaries of the aerosol particle layers in the free-troposphere (FT) and the atmospheric boundary layer (BL), we determined the top height of BL using the methodology described at Baars et al. (2008). The night-time BL over the measurement site ranged between 0.65 and 1.2 km while the mean top of the mixed layer height during daytime was at $2.0 \pm 0.3$ km (not shown here). The, rather low in altitude, PBL is suppressed by several limiting factors; 1) the frequent high pressure system in the region, 2) gravitational waves, 3) low wind speeds and 4) very dry air, which altogether limit convection. The gravitational waves define the horizontal transport of air and limit the growth of PBL to higher altitudes. In total, 844 FT aerosol particle layers were observed and 286 BL layers. To have a better insight of the time variation of the aerosol particle layers, Figure 4 presents the monthly geometrical and layer optical depth characteristics of BL and FT aerosol particles. Figure 4a corresponds to the centre of mass of the detected layers at the BL (red) and FT (green). While there is a very persistent and stable night-time BL at 1 km or so throughout the measurement year, the FT aerosol layers show seasonality. The FT particle layers extend to higher altitudes during the warmer months (April-August) and have a minimum height during November-December. Regarding their average geometrical depth (Fig. 4b), both BL and FT aerosol layers exhibit similar characteristics.

**The optical depths of BL and FT layers at 355 nm wavelength and their contribution to the total layer AOD are shown in Figures 4c-d. Similar conclusions are valid for AOD at 532 nm wavelength, which is not shown here but discussed in the manuscript. The optical depths were determined by integrating the layer aerosol extinction coefficient at 355 and 532 nm. For the first layer, where the overlap is incomplete, we assumed that the extinction coefficient value at the lowest trustworthy bin is representative for the values down to the surface to account for the incomplete overlap region. The highest layer AODs were measured during the summer months, and the lowest values during November and December for FT and February for BL. The mean (max) value of the total layer AOD amounts to $0.37 \pm 0.12$ (1.11) and $0.21 \pm 0.11$ (1.04) at 355 and 532 nm, respectively. These values are in line with previous studies, utilizing mainly sunphotometric observations at inland desert areas in the surrounding region (Ali et al., 2017; Eck et al., 2008). Moderate variations of the contribution of AOD in FT to the total layer AOD were observed for the investigated period (Fig. 4d). The contribution of the night-time FT layers to the total AOD was usually greater than that of the BL. Nevertheless, this behaviour was reversed from November to February. The lower total layer AODs in these months may be attributable to the absence of multiple FT layers, or to the lower surface wind speeds (which drive dust particles), during those months. There is a mesoscale phenomenon referred to as shamal conditions where northern to north-westerly winds are more intense between March to August compared to the rest of the year (Kutiel and Furman, 2003; Yu et al., 2016). Although the aforementioned values refer to night-time observations, on average the intra-day**



**variation in the region is moderate (Eck et al., 2008; Arola et al., 2013), and therefore comparable to this study.3.2 Intensive and extensive aerosol properties**

So far, we have examined the monthly variation of the aerosol layers over the measurement site in terms of their geometrical and AOD properties. In this section, we investigate the intensive (lidar ratios, linear particle depolarization ratios and Ångström exponents) and extensive (backscatter and extinction coefficients) aerosol properties of the retrieved aerosol particle layers

(Fig. 5). The backscatter and extinction coefficient values indicate occasional strong dust events. The dust events take place mainly between March and August when enhanced shamal conditions cause an increase in the probability of dust suspension and dust storms (Yu et al., 2016). Average $\beta$-values of $2.5 \pm 1.9$, $2.1 \pm 1.9$ and $1.6 \pm 1.6$ Mm$^{-1}$ sr$^{-1}$ for the 355, 532 and 1064 nm were observed, respectively. During strong dust events $\beta$-values up to 19.7 (18.5, 16.4) Mm$^{-1}$ sr$^{-1}$ at 355 (532, 1064) nm and $\alpha$-values of 800 (774) Mm$^{-1}$ at 355 (532) nm, were measured (not shown in the Figure). In general, the intensive optical

properties exhibit similar characteristics with little variation throughout the year apart from the period from mid-November to January. During the winter season increased LR values related to bigger Ångström exponents and lower linear particle depolarization values indicate a greater share of anthropogenic pollution in the aerosol particle mixture compared to other seasons.

Histograms of the aforementioned optical properties are shown in Figure 6. In the same figure, the statistical distribution is

also presented with box and whisker plots. For 40 (35) % of the cases, the LR at 355 (532) nm ranged between 35-45 sr while the second most frequent LR range was 45-55 (25-35) sr for the 355 (532) nm representing 27 (25) % of the cases. Furthermore, less than 12 % of the cases exhibited $\delta_p \geq 27$ % indicating the complexity of the aerosol type over the site; frequently a mixture of mineral dust (dominant aerosol) with anthropogenic and/or marine aerosol presence. This is also consistent with the backscatter-related Ångström exponent staying well below 0.8 in 71 % of the cases. In general, an average LR of $43 \pm 11$ sr

and $39 \pm 10$ sr was observed at 355 and 532, respectively. The mean $\delta_p$ was $15 \pm 6$ % for the 355 nm wavelength and $19 \pm 7$ % at 532 nm. A mean extinction-related Ångström exponent of $0.7 \pm 0.5$ between 355 and 532 nm was measured during the one-year period in UAE, similar to the value by Eck et al. (2008) based on sunphotometric observations in the greater area. Lastly, backscatter-related Ångström exponents at 355/532 and 532/1064 (not shown) were $0.6 \pm 0.4$ and $0.5 \pm 0.3$, respectively.

In order to reveal height-depended aerosol particle properties, we have further divided the atmosphere into 5 altitude ranges (0-1, 1-2, 2-3, 3-4 and >5 km) and grouped the aerosol properties contained in each altitude segment (Fig. 7). As expected, the $\beta$ and $\alpha$-coefficients decreased with increasing altitude. In contrast, LRs showed rather constant behavior up to 5 km suggesting similar aerosol mixtures throughout these altitude ranges. Interestingly, $\delta_p$ at 532 nm wavelength increased or remained constant with altitude except for aerosol layers above 5 km. This behavior was seen at 355 nm wavelength up to 2 km, but $\delta_p$

then decreased with altitude above 2 km. The most plausible explanation is that up to 2 km or so the night-time residual layers contain mixtures of mineral dust and anthropogenic pollution or/and marine aerosols resulting to lower linear particle



depolarization values. The mean relative humidity of these aerosol layers is much less than 60 % for 82 % of the cases hence hygroscopicity effects can be excluded. Ångström exponents increasing with altitude show the height-dependent nature of the aerosol size distribution (the higher the altitude the smaller the particles).

### 3.3 Optical properties of Arabian dust

To characterize the properties of mineral dust over the region, we have selected the top decile of linear particle depolarization values in the dataset. We discarded cases when the path of backward air mass trajectory passed over regions other than the Arabian Peninsula and the minimum height of the air mass over these regions was less than 3 km in altitude. The backward trajectories of the selected 46 cases are shown in Figure 8a and the characteristic optical properties and the aerosol type depended optical properties in Figures 8b-d for both 355 and 532 nm wavelengths, including a 95 % confidence ellipsoids. The mean values of all the aerosol particle optical properties are further reported in Table 1 along with literature values. The mean altitude of these layers was $1.8 \pm 0.9$ km; in 73 % of the cases the centre mass of the layer was located above 1 km excluding the stable and often well-mixed with anthropogenic or/and marine pollution, night-time BL. The retrieved dust aerosol properties over the region concerning the lidar ratios fluctuated between 35(34) and 55(54) sr with a mean value of $45 \pm 5$ ($42 \pm 5$) sr at 355 (532) nm. The values ranged between 22 (29) and 32 (35) % with an average value of $25 \pm 2$ ($31 \pm 2$) % for the linear particle depolarization ratio at 355 (532) nm and $0.3 \pm 0.2$ ($0.2 \pm 0.2$, $0.3 \pm 0.1$) for the extinction-related Ångström exponent (backscatter-related Ångström exponent at 355/532 and 532/1064). The ratio of LRs fluctuated between 1.0 and 1.2. Moreover, we report on the ratio of backscatter coefficients know as color ratio (CR) between 355/523, 355/1064 and 532/1064 wavelengths. This ratio is usually below 1 for aerosols and can be used in a simple aerosol/cloud detection scheme but dust particles show ratios above one which complicate this rather simple and straightforward relationship. Note that in the literature the CR is retrieved interchangeably either from smaller to bigger wavelength or the opposite. In this paper we calculated the CR as smaller to bigger wavelength.

To the authors' knowledge, four earlier studies have attempted to characterize the properties of dust originating from the Arabian Peninsula using the lidar technique, however, the full properties of dust were not characterised, particularly multi-wavelength optical properties and/or linear particle depolarization values have not been simultaneously defined. Müller et al. (2007), using lidar observations during INDOEX (Indian Ocean Experiment, Ramanathan et al., 2001) was first to stress the lower LR values of free-tropospheric dust when originating from the Arabian Peninsula compared to that from Northern Saharan. However, the long-range transported Arabian dust (aged) in their study suggest smaller LR values and greater Ångström exponents than the ones reported here (Table 1). Similar conclusions were found by Mamouri et al. (2013) and Nisantzi et al. (2015) whose studies show lower LR for the Arabian dust over a Mediterranean site in Cyprus than dust originating from the Saharan area, based on study cases (Table 1). A recent study by Hofer et al. (2017) using lidar observations in Tajikistan, Central Asia also report on Middle East dust optical properties and comparisons of those to Asian dust. To the





same direction, the study cases used over Dushanbe in Tajikistan show similar Arabian dust characteristics as in the present study (see Table 1).

A few limited studies are also available for the characterization of the Arabian dust properties using sunphotometric observations. Sunphotometric observations are column-integrated values which often include the contribution of BL aerosols and the contribution of non-dust aerosols (smoke, marine and anthropogenic aerosols). Nevertheless, Schuster et al. (2012) report a mean LR of 43 at 532 nm with a 39 to 43 range. On the contrary, Shin et al. (2018) result in higher LR values as $54 \pm 7$ at 440 nm and $37 \pm 4$ at 670 nm. The reported linear particle depolarization ratios are $0.21 \pm 0.03$ % at 440 nm and
$0.25 \pm 0.03$ % at 670 nm whereas the Ångström exponent $0.18 \pm 0.10$ at 440/870 nm.

Examining the reasons behind the different LR values in Arabian compared to African dust, previous studies related the optical characteristics to the chemical composition of the dust particles themselves. Numerous studies have analysed samples from various regions exploring the mineralogical composition of dust (Di Biagio et al., 2017; Egan G. Walter and Theodore, 1979; Querry, 1987; Sokolik and Toon, 1999). Schuster et al. (2012) linked the LR behaviour of dust to the percentage of illite in the
soil. The content of illite (K-rich argillaceous component of sedimentary rocks) in the dust defines the real refractive index which strongly influences LR. Since the real refractive index, which is determined by the mineralogical composition of dust defines the lidar ratio, an aerosol type parameter, it is expected that different dust types would exhibit different optical characteristics. Lower content of illite signifies lower LR compared to, for example, higher content illite in Saharan soils which result in the somewhat higher real refractive index. The refractive index of dust from Arabian peninsula is 1.48 and for Saharan
dust the corresponding value is 1.54 (Kim et al., 2011; Schuster et al., 2012). Towards this direction, we have collected two different dust samples from the area around the measurement site and further retrieved SEM images and performed elemental analysis (see Appendix A). Comparable to previous studies mentioned above, the fraction of K-rich argillaceous component of sedimentary rocks was well below 5.5 % in the collected dust samples.

## 4 Summary and conclusions

One-year of ground-based night-time lidar observations were analysed in synergy with backward air mass trajectories in order to characterize the seasonal variability of the background aerosol particle properties in a -heavy dust and anthropogenic polluted- region in the United Arab of Emirates (UAE). Our analysis suggests that aerosol particle populations over the UAE are sensitive to transport from Saudi Arabia, Iran, and Iraq but also from local sources. Two seasons exist in this area, summer and winter, where the main difference is the higher wind speeds between March-August compared to the rest of the year The
AOD was positively correlated with the season with maximum values being observed in the warmest months, June to August, resulting from the increased probability of dust suspension and dust storms. Multiple aerosol layers were present in the majority of identified cases, except during November-December; for 58 % of the cases the geometrical depth ranged between 0.4 and 0.8 km. The geometrical properties are determined by large scale pressure systems over the region as well as gravitational





waves introduced by local and regional topography. Regarding the optical properties, Ångström exponent values increased with altitude indicating the incapacity of bigger aerosols to reach higher up in the atmosphere. Lidar ratios were almost constant up to 5 km with a mean value of $43 \pm 11$ sr at 355 nm and $39 \pm 10$ sr at 532 nm. The linear particle depolarization, $\delta_p$, at 532 nm (355 nm), however, increased with altitude up to 3 km (2 km). The most probable explanation is that up to 2 km or so, night-time residual layers contain mixtures of mineral dust and urban-marine aerosols resulting in lower linear particle depolarization values. Higher up the linear particle depolarization decreases; the aerosol particles at higher altitudes are usually long-range transported and while aging in the atmosphere they become more spherical.

The Arabian Peninsula is a major contributor of airborne dust, yet very few studies have been made in order to characterize the pure dust optical properties of the region. To our knowledge this study is the first long-term one reporting the complete lidar-based optical characteristics of the Arabian dust. The FMI-Polly$^{XT}$ Raman lidar enabled the provision of lidar ratios and linear particle depolarization ratios at two wavelengths (355, 532 nm) giving us the possibility to answer to wavelength-dependent dust properties. The observed dust particle properties over the region regarding the lidar ratio amounted to $45 \pm 5$ sr at 355 nm and $42 \pm 5$ sr at 532 nm wavelength. Linear particle depolarization ratios of $25 \pm 2$ % ($31 \pm 2$ %) was observed at 355 (532) nm and $0.3 \pm 0.2$ ($0.2 \pm 0.2$) values was retrieved for the extinction-related Ångström exponent (backscatter-related Ångström exponent) at 355/532. The findings of this study suggest that the pure dust properties over the Middle East and western Asia, including the observation site, are comparable to those of African mineral dust regarding the linear particle depolarization ratios but not for the lidar ratios. The lower lidar ratio values are attributed to the different geochemical characteristics of soil with Arabian dust having lower K-rich values in the dust mixture, a component which determines the real refractive index of the dust. Implications of these findings propose that a universal lidar ratio for dust aerosol particles will lead to biased results, for example in satellite or ground-based extinction or aerosol typing retrievals as well as separation methods of a lidar signal to its aerosol components. In turn, all the aforementioned products are usually the basic input for advanced methodologies such as the retrieval of IN/CCN concentrations from lidar observations.

## 5 Data availability

The data used in this work are available upon request.

## 6 Author contribution

MF, MK and EG conceptualized and finalized the methodology. MF and MK were responsible for the lidar data and collection of the dust samples; RE and XS helped with the up keeping of the data and troubleshooting of instrument. JL analysed the dust samples. MF performed the data analysis and wrote the paper. All co-authors were involved in the paper editing, interpretation of the results and discussion of the manuscript.



## 7 Competing interests

The authors declare that they have no conflict of interest.

**Acknowledgements**

*This work was supported by the National Center of Meteorology, Abu Dhabi, UAE, under the UAE Research Program for Rain Enhancement Science. Hannele Korhonen received funding from the European Research Council (ERC) under the European Union's Horizon 2020 research and innovation programme under grant agreement No. 646857. Elina Giannakaki acknowledges the support of Academy of Finland (project no. 270108). The authors gratefully acknowledge the NOAA Air*
*Resources Laboratory (ARL) for the provision of the HYSPLIT transport and dispersion model used in this publication. The EM facility of SIB Labs at University of Eastern Finland is greatly acknowledged for providing their SEM and EDX equipment to the study. We would also like to thank Timo Anttila and Siddharth Tampi for providing onsite technical support.*

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

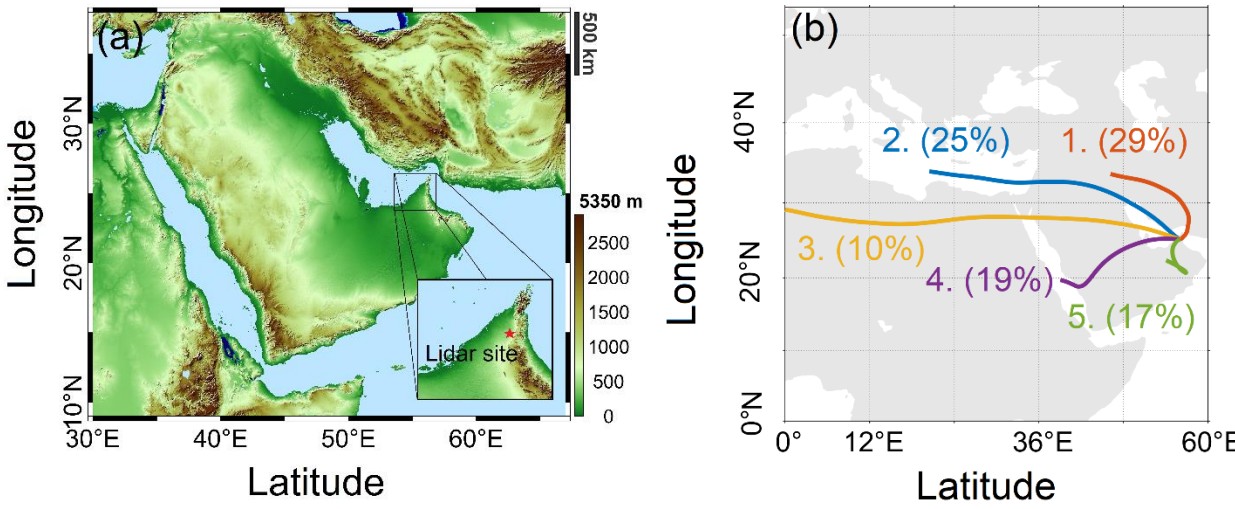


**Figure 1. (a) Digital Elevation Map (DEM) from the NASA Shuttle Radar Topographic Mission (*SRTM*) for the greater area under study. The colorbar values correspond to the altitude above sea level. The site location is shown at the bottom-right of this figure with a red star. (b) Cluster analysis of sources of the detected night-time aerosol layer in the region computed with HySPLIT over the course of the campaign period. Colored lines indicate the trajectory path and**
**the numbers show the percentage share of each trajectory path.**



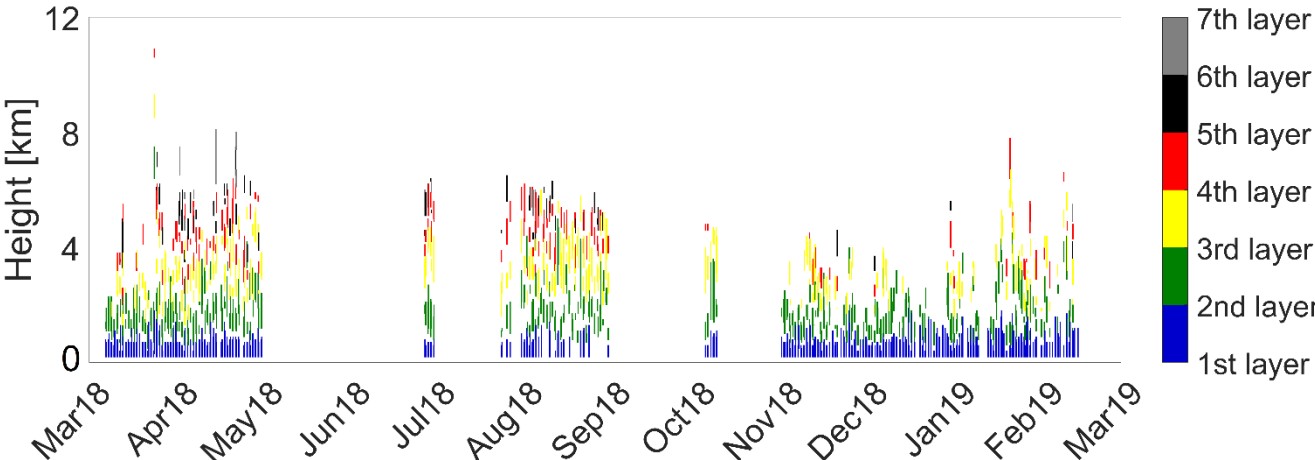

**Figure 2. Night-time geometrical boundaries of the aerosol layers observed between 6[th] March 2018 and 14[th] February**
**2019 at the measurement site in UAE. The color indicates the number of aerosol layers in the atmosphere. The gaps in**
**the dataset seen from May to August and between September and November were due to instrumental complications.**





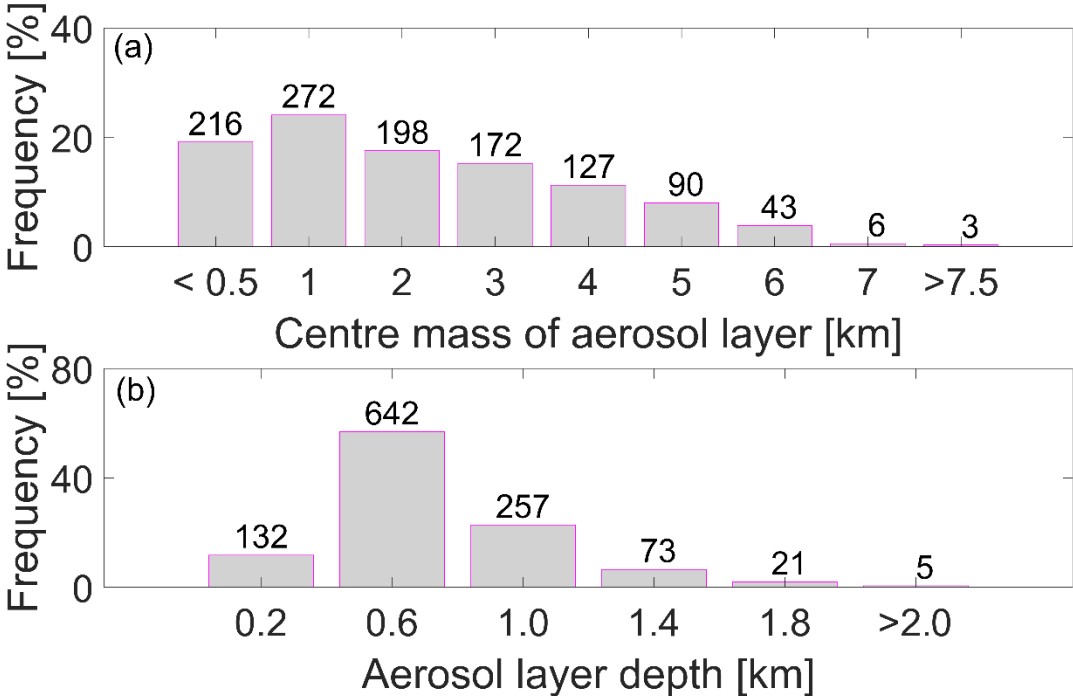


**Figure 3. Geometrical characteristics of the aerosol layers during the campaign period. (a) Frequency of the altitude of the centre of mass of the aerosol layers. The width of each bin 1 km apart from the first and last bins. (b) Frequency of the geometrical depth of the aerosol layers. The width of each bin is 0.4 km apart from the last bin. The numbers on top of the bars indicates the amount of cases included in the bin.**




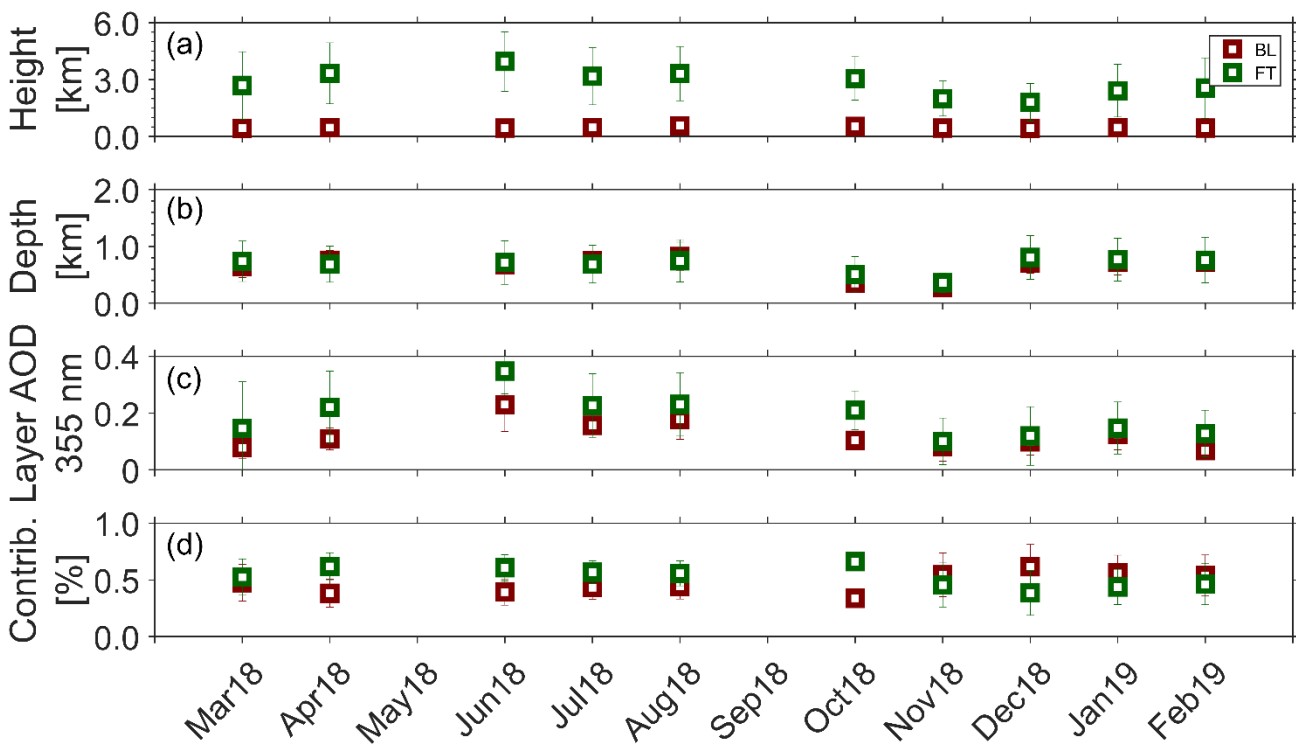

**Figure 4. Geometrical characteristics and optical properties of the detected night-time aerosol layers in the boundary
layer (BL; in red) and free-troposphere (FT; in green). (a) Centre of mass height of the aerosol layers. (b) Geometrical
depth of the detected layers. (c) Layer aerosol optical depths (AOD) at 355. (d) Contribution of BL (red) and FT (green)
aerosol layers to the total AOD.**



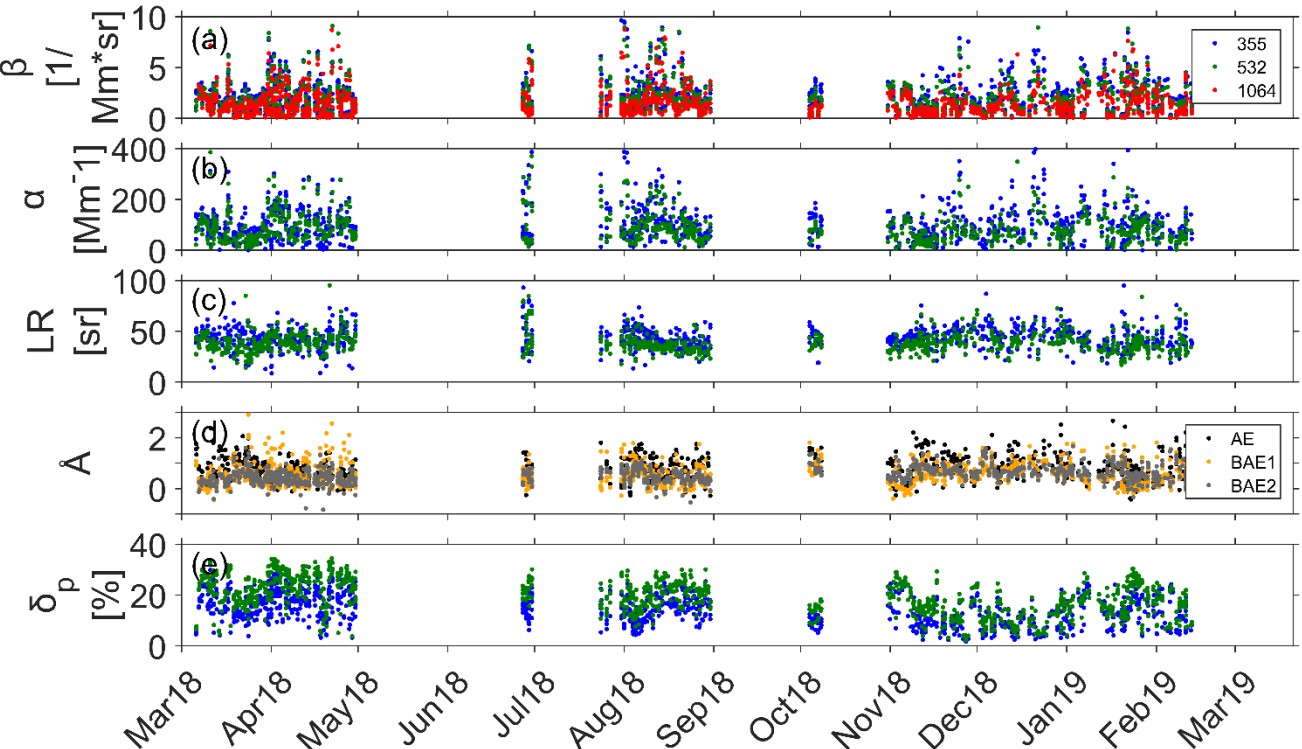

**Figure 5. Intensive and extensive aerosol properties at different wavelengths (355 nm in blue, 532 nm is green and 1064 nm in red). (a) Backscatter coefficient. (b) Extinction coefficient. (c) Lidar ratio. (d) Ångström exponents (Å) where the extinction-related Ångström exponent (AE) is marked with black dots, the backscatter -related Ångström exponent at 355/532 (BAE1) with orange and the backscatter-related Ångström exponent at 532/1064 (BAE2) with grey. (e) Linear particle depolarization ratio ($\delta_P$).**





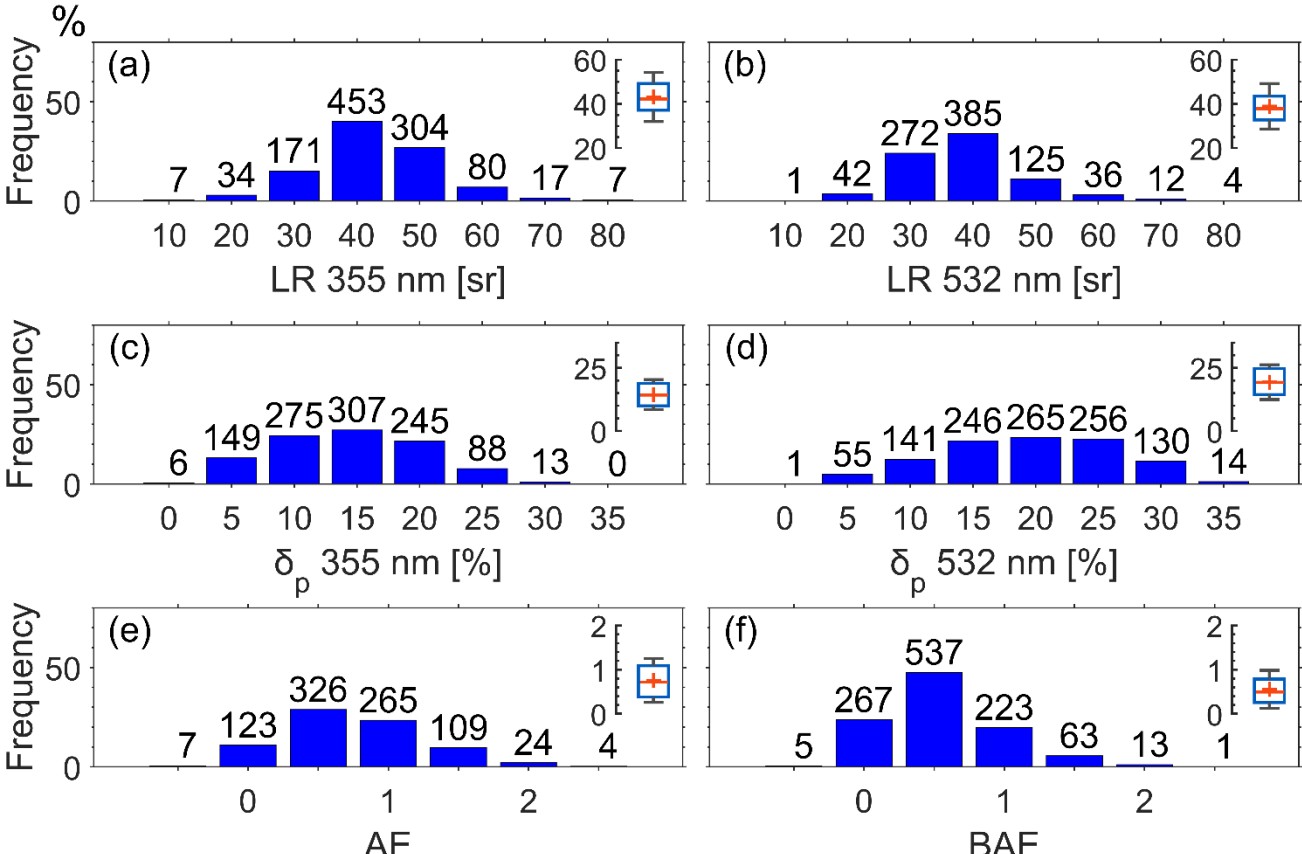

**Figure 6. Frequency distribution of: (a) Lidar ratio at 355 nm with bin width of 10 sr. (b) Lidar ratio at 532 nm with bin width of 10 sr. (c) Linear particle depolarization ratio at 355 nm with bin width of 5 %. (d) Linear particle depolarization ratio at 532 nm with bin width of 5 %. (e) Extinction-related Ångström exponent (AE) at 355/532 with bin width of 0.5. (f) Backscatter-related Ångström exponent (BAE) at 355/532 with bin width of 0.5. Box and whisker plots are also presented where cross is the mean value, horizontal line is the median, boxes are the 25 and 75 % percentiles respectively, and whiskers represent the one standard deviation. The numbers above the bars indicate the amount of cases fallen in the bin.**





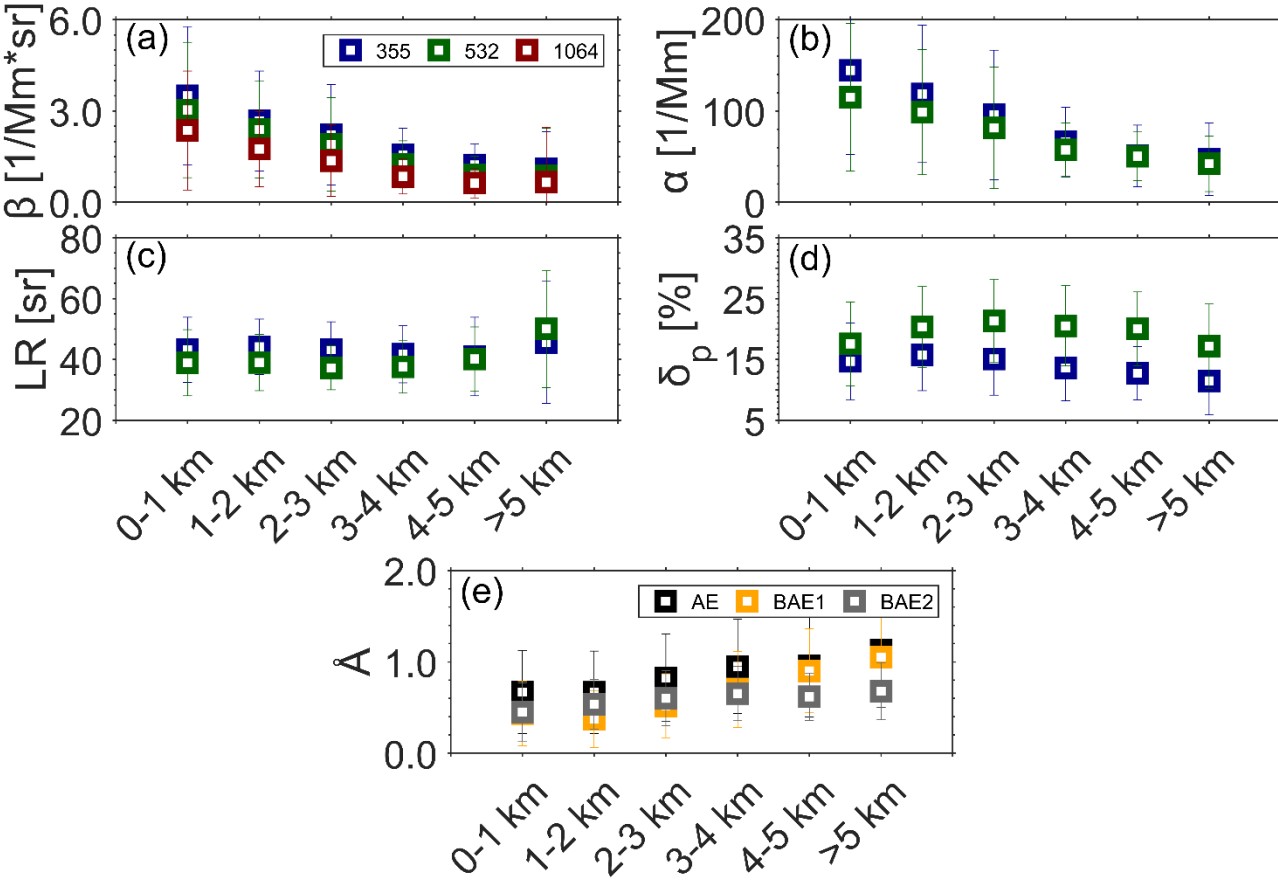

**Figure 7. Height-dependent aerosol properties for 0-1, 1-2, 2-3, 3-4 and >5 km altitude. (a) Backscatter coefficient at 355 nm (blue), 532 nm (green) and 1064 nm (red). (b) Extinction coefficient at 355 nm and 532 nm. (c) Lidar ratio at 355 nm and 532 nm. (d) Linear particle depolarization ratio at 355 nm and 532 nm. (e) Extinction-related Ångström exponent (AE) at 355/532, Backscatter-related Ångström exponent (BAE1) at 355/532 and Backscatter-related Ångström exponent (BAE2) at 532/1064.**



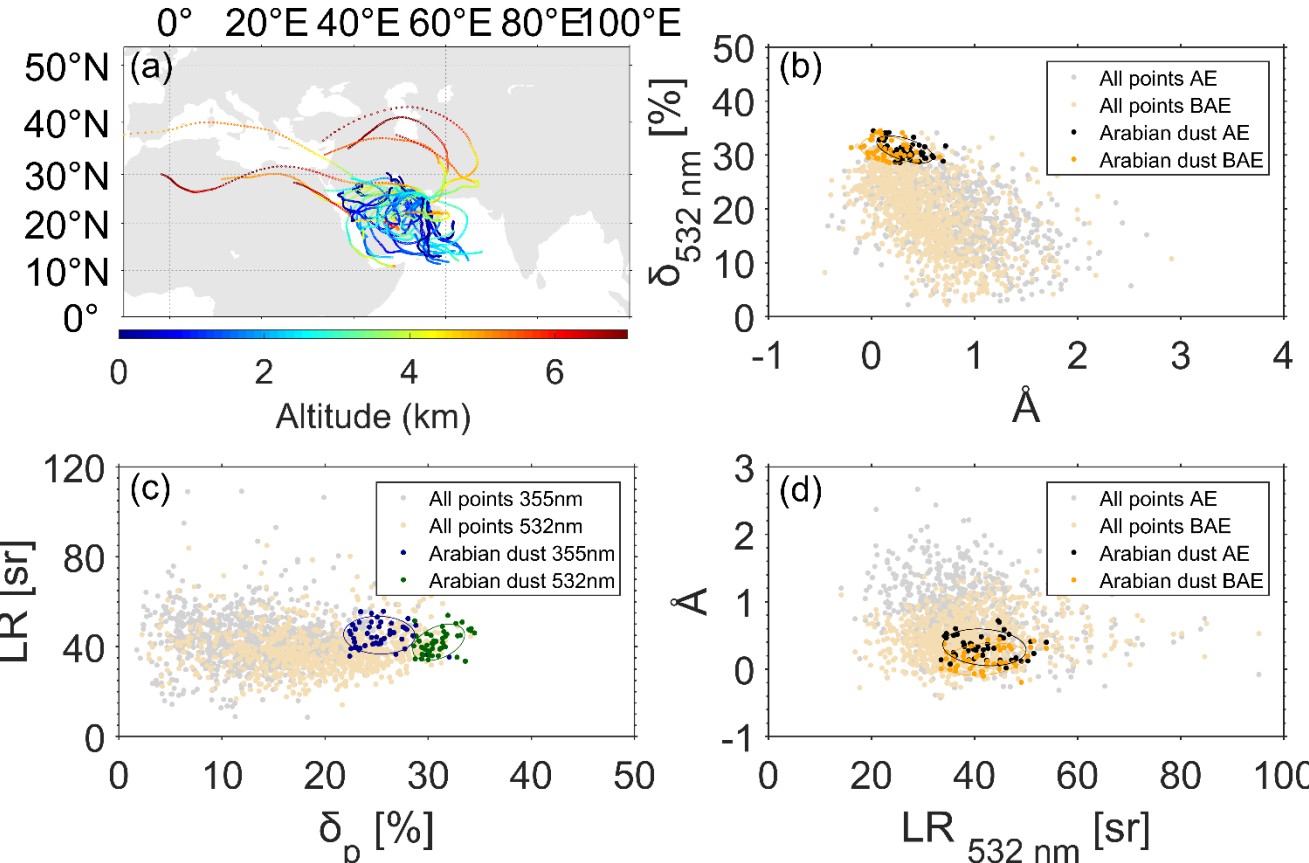


**Figure 8. (a) Backward air mass trajectories of all the cases considered for the characterization of the Arabian dust properties. (b) Ångström exponent (Å) versus linear particle depolarization ($\delta_p$) at 532 nm. (c) Linear particle depolarization ($\delta_p$) versus lidar ratio (LR). (d) Lidar ratio (LR) at 532 nm versus Ångström exponents (Å). The Ångström exponent plots at b and d panels indicate the extinction-related (AE) and backscatter-related (BAE) exponents at 355/532. The ellipsoids in panels b-d were drawn considering a 95% confidence for the set of data points.**





**Table 1: Aerosol particle properties of the Arabian dust and comparison to previous studies. Both 355 and 532 nm wavelengths are reported in terms of their lidar ratio (LR), linear particle depolarization ($\delta_p$) and the ratio of theirs lidar ratios. Combination of Ångström exponents both from the extinction (AE) and backscatter (BAE) coefficients along with the color ratios (CR) are shown due to the multi-wavelength capability of the lidar instrument. The numbers in the brackets show the range of values found for each optical property.**


| Property | LR 355nm [sr] | LR 532nm [sr] | $\delta_p$ 355nm [%] | $\delta_p$ 532nm [%] | AE | BAE 355/532 | BAE 532/1064 | CR 355/1064 | CR 532/1064 | CR 355/532 | LR355/ LR532 |
|---|---|---|---|---|---|---|---|---|---|---|---|
| Müller et al. (2007) | 38 ± 5 | 38 ± 5 | - | - | 0.6 ± 0.3 | - | 1.1 ± 0.4 | - | - | - | ~1 |
| Mamouri et al. (2013) | - | 34 ± 7 to 39 ± 5 | - | (28-35) | - | - | - | - | - | - | - |
| Nisantzi et al. (2015) | - | 41 ± 4 (33-48) | - | - | - | - | - | - | - | - | - |
| Hofer et al. (2017) | 42 ± 3 | 36 ± 2 | 18 ± 2 | 31 ± 1 | 0.4 ± 0.2 | 0.0 ± 0.2 | 0.1 ± 0.0 | - | - | - | - |
| This study | 45 ± 5 (35-55) | 42 ± 5 (34-54) | 25 ± 2 (22-32) | 31 ± 2 (28-35) | 0.3 ± 0.2 (0.0-0.7) | 0.2 ± 0.2 (-0.2-0.7) | 0.3 ± 0.1 (0.1-0.6) | 1.4 ± 0.2 (1.1-2.0) | 1.3 ± 0.1 (1.1-1.5) | 1.1±0.1 (0.9-1.3) | 1.1 ± 0.1 (0.9-1.3) |






## Appendix A

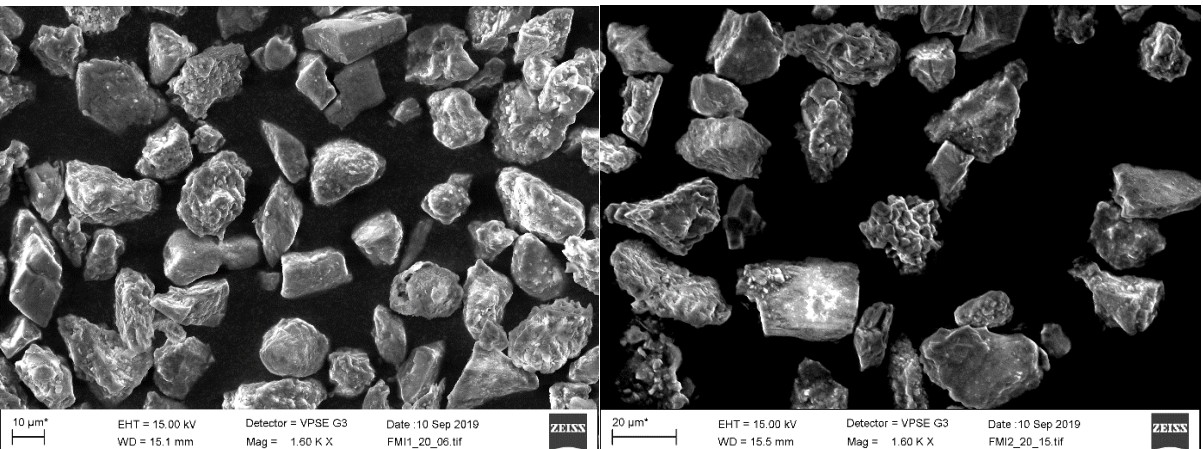

**Figure B1: SEM pictures of the two dust samples. Sample 1 on the left and sample 2 on the right.**


**Table C1. Chemical composition of the two dust samples. The numbers show the minimum/maximum value of the elements found in the sample.**

| Weight % | Sample 1 | Sample 2 |
|---|---|---|
| O | 45-82 - 64.49 | 50.79 - 59.25 |
| F | - | 6.17 |
| Na | 0.48 - 3.38 | 0.55 -2.10 |
| Mg | 2.68 - 6.30 | 2.66 - 10.04 |
| Al | 1.48 - 10.88 | 1.79 - 3.38 |
| Si | 5.45 - 23.89 | 6.09 - 35.87 |
| S | 0.30 - 15.87 | 0.21 - 1.93 |
| K | 0.30 - 5.39 | 0.21 - 0.74 |
| Ca | 6.43 - 33.55 | 4.04 - 32.33 |
| Ti | - | 0.83 - 35.88 |
| Fe | 1.20 - 12.33 | 2.08 - 5.62 |
| Ba | 0.90 | - |
| Cl | 0.20 | - |