# Peer review of "Optical and geometrical aerosol particle properties over the United Arab Emirates"

_Atmospheric Chemistry and Physics, 2020_

## Referee Comment (RC1) · Anonymous Referee #3 · 9 Apr 2020

REVIEW

Optical and geometrical aerosol particle properties over the United Arab Emirates by Maria Filioglou et al.

GENERAL COMMENT

The paper deals with a relevant topic within the scope of Atmospheric Chemistry and Physics. The manuscript is well written and structured in a rather convenient way. Although the period covered is short, specially having in mind the gaps due to instrumental problems, the data set is very valuable having in mind the quality and variety of variables analyzed and the scarcity of atmospheric aerosol vertical profiling in the study area. Searching similarities and differences with North African mineral dust aerosols is

worthy, and the authors do this combining remote sensing vertical profiling and some lab work on collected samples. The characterization provided is very comprehensive and at the same time presents results in a very concise way. The paper is appropriate for ACP but in the present stage it requires some revisions in aspects I detail in the following section.

PARTICULAR COMMENTS

There are some issues related to the analysis of the atmospheric vertical structure. Thus, the results on Free Troposphere, FT, and Boundary Layer, BL, are a little bit surprising. The depth of the FT is according to figure 4 is usually rather low, around 1 km, similar to that of BL, something that it is not coherent with the definition of FT. On the other hand, the authors must explain the procedure concerning the computation of the center of mass of FT and BL, justifying if they consider this as a geometrical variable or if it is computed having in mind the variability of density with height, otherwise the results can't be interpreted. Concerning the nighttime results for BL height it is rather surprising the range of values obtained, too high for representing the top of the stable boundary layer. So this part requires some discussion and explanation of the procedures applied.

Figure 5 describes the evolution of extensive and intensive aerosol variables. The problem is that the authors do not clarify the meaning of the backscatter and extinction coefficient presented in figure a and b. Are they average values?, in such case more info like standard deviation would be necessary. Are they representative values: max? This requires additional information.

On the other hand, Figure 5 uses rather raw scales and in order to support some of the discussions on how the different variables change and offers some typing insights it is necessary a better representation. In this sense, Figure 8 showing the relationships of pairs of intensive variables offer some insight on the typing comments offered by the authors, although the spread of data in the selected scatter plots hardly support some

statements linking particle depolarization with lidar ratio or Angström exponent with lidar ratio. Only the figure with the scatter plot of depolarization ratio versus Angström exponent presents some dependence between these variables. So the authors must improve the way they present their analyses of this intensive variables.

---

## Referee Comment (RC2) · Anonymous Referee #1 · 22 May 2020

The paper of Filioglou et al., presents the geometrical and optical properties of the Arabian dust particles based on ground-based observations from a multi-wavelength Raman lidar instrument. The manuscript is well written and structured and the main results of the study are very interesting. I recommend the publication of the manuscript after some revisions, considering the following comments.

1) The measuring period covers an almost one year of observations (from March 2018 to February 2019), with two measuring gaps during May to August and September to November, due to instrumental problems. Thus, the term "long-term observations" used by the authors, should be replaced through the manuscript with the "one year observations".

2) In the introduction part, the authors should discuss about the threshold values of

the intensive optical properties (lidar ratio, depolarization ratio) used in existing typing schemes for dust particles within EARLINET for example. Stations within EARLINET, are affected mostly from the African dust, so the references clusters are attributed to properties connected with these particles. But, what about stations e.g. Cyprus, affected by both the African mineral dust and the Arabian dust. This discussion would strengthen the claim of the authors that "a universal lidar ratio for dust aerosol particles will lead to biased results".

3) In the processing part, the authors should discuss more the automatic detection of the aerosol particle layers. Do they use a minimum layer thickness threshold (Figure 3 indicates that they did not). How do they define the first detected layer. Is this the PBL top? Please explain. In the manuscript you state that " there is a very persistent and stable night-time BL at 1 km or so throughout the measurement year". Is this the first layer presented in Fig. 2.

4) Figure 4 and Figure 5, present inconsistent retrievals for June 2018. Figure 4 presents geometrical properties for June 2018, while Figure 5 presents missing data. Please correct the figures accordingly.

5) Figure 4b, is a bit misleading. As it is shown it gives the impression that the FT has a certain depth equal to the PBL depth. Please modify.

6) Figure 7. Maybe the authors can provide a different approach for these plots. The division of the atmosphere into 5 altitude ranges (0-1,1-2,2-3,3-4 and >5) is a bit supressed. Maybe you could provide the information, based on the division of the atmosphere in regions, PBL, FT.

7) Figure 8. Authors should discuss more about the correlation of the presented properties of the Arabian dust. Can they conclude about the correlation between LR and $\delta$? Or between the other properties?

8) Last paragraph of 3.3. The authors discuss about the possible differences between

[Figure]
* * *
Interactive
comment

African dust and the Arabian dust, analyzing two dust samples. However, they provide limited information about these two samples. Why are they interesting? They are linked to particular transported aerosol load? What about the lidar properties obtained during these periods of sampling? These are issues that the authors should address, so as the reader to understand the connection with the current analysis.

9) Line 295. Please provide a reference to strengthen the statement.

---

## Author Comment (AC1) · 11 Jun 2020

*The authors would like to thank the reviewer for his/her valuable comments and suggestions. We have modified the manuscript with the proposed changes along with step by step answers to the suggestions. Please note that changes have been highlighted (in bold or 'track changes') in the manuscript and the corresponding answers to the reviewer by text below. The original comments are presented in bold letters.*

**Comment_1: There are some issues related to the analysis of the atmospheric vertical structure. Thus, the results on Free Troposphere, FT, and Boundary Layer, BL, are a little bit surprising. The depth of the FT is according to figure 4 is usually rather low, around 1km, similar to that of BL, something that it is not coherent with the definition of FT. On the other hand, the authors must explain the procedure concerning the computation of the center of mass of FT and BL, justifying if they consider this as a geometrical variable or if it is computed having in mind the variability of density with height, other-wise the results can't be interpreted.**

To rule out the possibility of misunderstanding we clarify here that Figure 4 and the related text discusses about FT aerosol layer and BL aerosol layer depths and other related properties, not the FT or BL depths even though BL aerosol layer gives an indication of the BL depth. This has been clarified in the updated manuscript to avoid further misunderstanding.

Figure 4 shows monthly mean nighttime variations of FT aerosol layers and BL aerosol layers regarding their a) altitude b) geometrical depth c) optical depth and d) their contribution to the total layer AOD. The mean nighttime height of the FT aerosol layers is presented in Fig 4a and corresponds to 2.8 ± 1.4 km for the whole period where monthly variations can be seen in the figure itself. The mean monthly variation of the aerosol layers are reported through their center of mass which accounts for the variability of the density within each layer. The geometrical thickness of these aerosol layers is shown in Fig. 4b. This depth is computed from the boundary of the top/bottom height of the aerosol layer. The top/bottom height of the aerosol layers is detected using the second derivative of the backscatter profiles. We aimed to report here the real aerosol geometrical boundaries and not weight the reported depth values by accounting the variability of density with height. The latter case is already accounted as the geometrical depth is larger than the variation of the height of center of mass.

We have modified the text and labels referring to Figure 4 where possible to clarify that we refer to Free-tropospheric aerosol layers and not the properties of free troposphere itself.

*"The geometrical depth is calculated from the aerosol layer boundaries (top/bottom) in which the subtraction of these boundaries result to the actual geometrical thickness of the corresponding aerosol layers."*

**Comment_2: Concerning the nighttime results for BL height it is rather surprising the range of values obtained, too high for representing the top of the stable boundary layer. So this part requires some discussion and explanation of the procedures applied.**

The BL top height was retrieved using the methodology described by Baars et al. (2008) in which the wavelet covariance transform method (wct) shows a local maximum at BL top. For the BL during nighttime, the residual layer and the stable BL are located below that height. Above the transition zone, FT layers may be present. In this sense, the reported nighttime BL height range (0.35 to 1.2 km) depict

not necessarily the stable BL but include the residual layer(s) as well. Nevertheless, a previous study in the coastal area near the site show the formation of a stable marine BL at 500-800 m in altitude (Reid et al., 2008).

*"It should be noted that the observed lidar PBL height is expected to depict, apart from any mechanically driven layer during the stable and transition periods, the top of the residual layer from the previous day."*

Reid, J. S., et al. (2008), An overview of UAE[2] flight operations: Observations of summertime atmospheric thermodynamic and aerosol profiles of the southern Arabian Gulf, *J. Geophys. Res.*, 113, D14213, doi:[10.1029/2007JD009435](10.1029/2007JD009435).

**Comment 3: Figure 5 describes the evolution of extensive and intensive aerosol variables. The problem is that the authors do not clarify the meaning of the backscatter and extinction coefficient presented in figure a and b. Are they average values? in such case more info like standard deviation would be necessary. Are they representative values: max? This requires additional information.**

Thank you for your comment. Indeed the reported values in Figure 5 are mean values of the optical parameters for each detected aerosol layer.

We have added the standard deviation in Figure 5 and changed the text in the manuscript to indicate it so.

**Commnet_4: On the other hand, Figure 5 uses rather raw scales and in order to support some of the discussions on how the different variables change and offers some typing insights it is necessary a better representation. In this sense, Figure 8 showing the relationships of pairs of intensive variables offer some insight on the typing comments offered by the authors, although the spread of data in the selected scatter plots hardly support some statements linking particle depolarization with lidar ratio or Angström exponent with lidar ratio. Only the figure with the scatter plot of depolarization ratio versus Angström exponent presents some dependence between these variables. So the authors must improve the way they present their analyses of this intensive variables.**

Figures 5 to 7 report aerosol properties derived from the lidar measurements using three different approaches. Figure 5 shows the time series of the intensive and extensive aerosol properties and through this figure we intend to discuss the seasonal variation of the optical properties, if any, and connect them with meteorological conditions and anthropogenic activities in the region. This overall picture is complemented by Figures 6 and 7 which look further into the aerosol properties. In Figure 6, histograms present the most frequent values regarding the linear depolarization ratio, lidar ratio and Angstrom exponents over the site raising a climatological value for the site, while Figure 7 shows the vertical distribution and validity of these optical properties with height. We do not intend to perform a case-by-case aerosol typing as this task is rather challenging and to some extend limited due to the lack of accurate information from auxiliary observations and/or models. For example, we have seen that aerosols over the site are often a mixture of anthropogenic and/or marine aerosols. With the use of backward trajectories, satellite observations and aerosol composition derived from models we could

classify these aerosol layers, to some extent, but our intention here is not this. As detail aerosol characterization at ground-level will be answered by our in-situ measurements, we focus on the retrieval of the Arabian dust properties. To this direction, Figure 8 shows the dependence of some of the optical properties for the Arabian dust particles. In Figure 8 apart from the Arabian dust layers, the rest of the dataset is also presented. This shows the great variability of the lidar-derived optical properties during the year-long campaign period hence the variability in the aerosol types found over the measurement site. By using the methodology described in the paper we were able to characterize the Arabian dust optical properties. Such was possible due to the high quality of the retrieved aerosol properties and also due to the absence of volcanic aerosols in these altitudes which enabled to connect the high linear depol. ratios with the specific aerosol type with confidence. The relationships of the optical parameters used in Figure 8 are common pairs reported in the lidar literature (e.g Groß et al., 2013 & 2015) and such relationships are currently used in aerosol typing methodologies (e.g the NATALI code, Nicolae et al., 2018). Furthermore for the various optical parameters, the reported range of values for the Arabian dust cases fall within the uncertainty of the measurement themselves (including systematic and statistical errors) presenting the best possible option at the moment.

For better clarity we didn't include the error bars in Figure 8 but we include it here in the response. We have also included a paragraph discussing about the rest of the dots in Figs 8b-d.

[Figure]

Groß, S., Esselborn, M., Weinzierl, B., Wirth, M., Fix, A., and Petzold, A.: Aerosol classification by airborne high spectral resolution lidar observations, Atmos. Chem. Phys., 13, 2487–2505, https://doi.org/10.5194/acp-13-2487-2013, 2013.

Groß, S., Freudenthaler, V., Schepanski, K., Toledano, C., Schäfler, A., Ansmann, A., and Weinzierl, B.: Optical properties of long-range transported Saharan dust over Barbados as measured by dual-wavelength depolarization Raman lidar measurements, Atmos. Chem. Phys., 15, 11067–11080, https://doi.org/10.5194/acp-15-11067-2015, 2015.

Nicolae, D., Vasilescu, J., Talianu, C., Binietoglou, I., Nicolae, V., Andrei, S., and Antonescu, B.: A neural network aerosol-typing algorithm based on lidar data, Atmos. Chem. Phys., 18, 14511–14537, https://doi.org/10.5194/acp-18-14511-2018, 2018.

---

## Author Comment (AC2) · 11 Jun 2020

*The authors would like to thank the reviewer for his/her valuable comments and suggestions. We have modified the manuscript with the proposed changes along with step by step answers to the suggestions. Please note that changes have been highlighted (in bold or 'track changes') in the manuscript and the corresponding answers to the reviewer by text below. The original comments are presented in bold letters.*

**Comment_1: The measuring period covers an almost one year of observations (from March 2018 to February 2019), with two measuring gaps during May to August and September to November, due to instrumental problems. Thus, the term "long-term observations" used by the authors, should be replaced through the manuscript with the "one year observations".**

Thank you for your comment. Indeed the challenging conditions at UAE did not allow us to have observations for the full time of the campaign. The manuscript has been updated according to the reviewer's suggestion.

**Comment_2: In the introduction part, the authors should discuss about the threshold values of the intensive optical properties (lidar ratio, depolarization ratio) used in existing typing schemes for dust particles within EARLINET for example. Stations within EARLINET, are affected mostly from the African dust, so the references clusters are attributed to properties connected with these particles. But, what about stations e.g. Cyprus, affected by both the African mineral dust and the Arabian dust. This discussion would strengthen the claim of the authors that "a universal lidar ratio for dust aerosol particles will lead to biased results".**

We have added a paragraph in the introduction discussing the lidar ratios used in EARLINET community and the CALIPSO retrievals. A few sentences were also added at the conclusion section. The additions can be also found below:

*"The lidar ratio is a parameter commonly used in lidar based aerosol typing algorithms to classify the particles within an atmospheric layer (Nicolae et al., 2018; Papagiannopoulos et al., 2018). This parameter is also critical for elastic lidar retrievals and separation techniques (e.g Giannakaki et al., 2020 and references therein). Within the European Aerosol Research Lidar Network (EARLINET, Pappalardo et al., 2014), stations are typically affected from dust outbreaks originating from the Western Saharan region. Amiridis et al. (2013) retrieved Saharan dust lidar ratios at 532 nm of 58 ± 8 sr, while a mean (range) value considering all EARLINET stations is 51 ± 10 sr (30-80 sr), at the same wavelength (Papagiannopoulos et al., 2016). Currently, a mean value of 55 sr is used for dust related applications (e.g Tesche et al 2009). On the contrary, the aerosol classification scheme from the satellite based lidar onboard CALIPSO (Vaughan et al., 2009) uses a lidar ratio for pure dust of 44 ± 9 sr (Kim et al., 2018). Nevertheless, neither approaches consider the origins of the dust which translates into different optical characteristics (African or Middle East)."*

And

*"This becomes more evident in stations where they are the receptors of both dust types and the selection of adequate dust optical parameters is important for further analysis."*

**Comment 3: In the processing part, the authors should discuss more the automatic detection of the aerosol particle layers. Do they use a minimum layer thickness threshold (Figure 3 indicates that they did not). How do they define the first detected layer. Is this the PBL top? Please explain. In the manuscript you state that "there is a very persistent and stable night-time BL at 1 km or so throughout the measurement year". Is this the first layer presented in Fig. 2.**

The aerosol layer detection uses the second derivative of the 1064 nm channel (532 nm in the absence of it and lastly 355 nm, if nothing else is available) to detect zero crossings in the signal. Because this method suffers from signal noise, we first smooth the signal and then retrieve the layer boundaries. We do use a minimum layer thickness threshold of 50 m and we also use minimum thresholds for the mean backscatter values within the aerosol layers (0.25, 0.10 and 0.05 $Mm^{-1}$ $sr^{-1}$ for the 355, 532 and 1064nm, respectively). So after the initial detection of the layers, we discard those with less than 50 m in depth and those who have low backscatter values as the statistical errors become significant. A more detailed description has been added to the corresponding section (Sect. 2.3).

The first detected layer comes from the above methodology were the base is always set to 0 m (starts from the ground) and this is the layer shown in Fig 2. It doesn't necessarily mean that the top boundary of this first layer is the PBL top (although in almost all cases it is), as the PBL is retrieved for each profile following the methodology described in Baars et al., 2008. The WCT (wavelet covariance transformation) method looks for a steep decrease in the backscattered signal at the top-height of this layer. At that point the function takes a local maximum. It is preferred for the PBL detection over the gradient method as it is less exposed to signal noise and it doesn't require vertical smoothing. Therefore, the WCT method was preferred over the gradient method for the BL top height detection. Then, the separation between BL and FT is straightforward and can be done by accounting aerosol layers falling below the BL top height (BL aerosol layers) or not (FT aerosol layers). We acknowledge here that the BL detection is not necessary the stable night-time BL layer but it often includes the residual aerosol layer from the previous day but this falls within the capabilities of any existing lidar and depends on the atmospheric conditions. Some clarification about the PBL was also added in Sect. 3.1

**Comment_4: Figure 4 and Figure 5, present inconsistent retrievals for June 2018. Figure 4 presents geometrical properties for June 2018, while Figure 5 presents missing data. Please correct the figures accordingly.**

Thank you for your comment. To clear out any confusion, Figure 4 presents mean monthly values of geometrical and AOD properties including all the available retrieved aerosol profiles separated by the height (BL vs FT) while Figure 5 presents the time series of these optical properties. Therefore, the x-axis on Figure 5 presents mean values of each retrieved aerosol layer for each day. The availability of lidar measurements during June and July was limited and only at the very end of each of these two months aerosol profiles were retrieved. Note that the marking of the month in the x-axis in Fig. 5 corresponds to the first day of that month.

**Comment_5: Figure 4b, is a bit misleading. As it is shown it gives the impression that the FT has a certain depth equal to the PBL depth. Please modify.**

We have now updated the legend of Figure 4 from BL/FT to BL/FT aerosol layers.

**Comment_6: Figure 7. Maybe the authors can provide a different approach for these plots. The division of the atmosphere into 5 altitude ranges (0-1,1-2,2-3,3-4 and >5) is a bit suppressed. Maybe you could provide the information, based on the division of the atmosphere in regions, PBL, FT.**

Thank you for the suggestion. With Figure 7 we aimed to reveal height-dependent differences in the mean aerosol optical properties and further deepen our understanding of the aerosol mixtures in the region. It nicely captures the higher linear depolarization ratios at higher altitudes compared to the aerosol layers below 1 km helping us to connect properties of local dust with the rest aerosol types present. At most times dust was found as a mixture of anthropogenic pollution or marine contribution in the area considering aerosol layers below 1 km. For clarification purposes, we attach here the mean optical properties, as suggested, separated into three aerosol layer groups: all (this is the same as the inner plot in Figure 6), PBL and FT. As it can be clearly seen the behavior of the different altitudes is now smoothed and no further information can be distinguish between BL and FT aerosol optical properties. To some extent, this graph is a bit misleading showing higher Angstrom exponents and lower particle depolarization ratios for the FT category which is not necessarily true as already presented in Fig. 7 (for the lower free-troposphere).

[Figure]

**Comment_7: Figure 8. Authors should discuss more about the correlation of the presented properties of the Arabian dust. Can they conclude about the correlation between LR and δ? Or between the other properties?**

We have now added more discussion related to the optical properties of Arabian dust and their difference with African dust. The text has been updated considering the suggestion above (Sect.3.3).

*"Figures 8b-d also present the rest of the available aerosol layer optical properties apart from the ones defined as Arabian dust. We assume that cases with the highest linear depolarization ratios represent the pure Arabian dust optical properties since no volcanic aerosols are present over the measurement site at the given period. It is also evident form the scatter plots that the aerosol sources and types are varying thus no detailed conclusions can be made for the full data set except the ones mentioned in previous sections."*

And

*"All in all, the Arabian dust optical properties show close to 0 Ångström exponents and high linear depolarization ratios, characteristics that are similar to Saharan dust. Dissimilarly, Arabian dust has lower LRs and these LRs are almost equal at 355 nm and 532 nm. Another difference is the CRs which are well above 1 ranging from 1.1 up to 2.0 depending of the wavelength selection"*

**Comment_8: Last paragraph of 3.3. The authors discuss about the possible differences between African dust and the Arabian dust, analyzing two dust samples. However, they provide limited information about these two samples. Why are they interesting? They are linked to particular transported aerosol load? What about the lidar properties obtained during these periods of sampling? These are issues that the authors should address, so as the reader to understand the connection with the current analysis.**

We have added text accordingly. The collected samples are not linked to a particular transported aerosol load. They are normal soil samples collected from two different regions around the site. The collection was made regarding the different observed soil color and we also accounted regional discrepancies by taking samples closer and further away from the measurement site. We have added a few more sentences explaining the dust sample part in Sect 2.4 where we introduce the sampling methodology.

*"The samples were dry collected accounting both microphysical (color of the soil) and regional discrepancies; one sample was close to the observation site and the other one a few tens of kilometers away."*

And

*"Moreover, the study of Di Biaggio et al. (2019) report that the imaginary part of the refractive index in dust samples originating from Saudi Arabia score less than that of African dust, presenting a lower absorbing efficiency compared to African dust. The difference is attributed to the content of iron oxides in the dust"*

And

*" Regarding the absorption properties of dust, it has been found that dust optical properties are more correlated to the fraction of iron oxides than the iron content itself. Nevertheless, the iron content in the collected dust samples was lower than that recorded in African dust (Di Biaggio et al. 2019)."*

**Commnet_9: Line 295. Please provide a reference to strengthen the statement.**

The references for the connection of refractive indexes and the amount of illite in the soil come in the next sentence.

---

## Editor Comment (EC1) · Stelios Kazadzis (Editor) · 14 Jun 2020

One of the questions posed in the review of this work was:

are all the uncertainties presented, come from the mean and standard deviation of the parameters and why not systematic uncertainties (Lidar related calibration etc.) were not included ?

with an answer:

Regarding the uncertainties presented, the reported standard deviations include the uncertainties of the lidar measurements themselves.

I think the way this is answered needs a bit more clarification. For example LR mea-

surements presented in fig. 7

Are the LR bars the result of both error propagation of the backscatter and the extinction plus the statistical uncertainty ? how much are those individual uncertainties ?
* * *

---

## Author Comment (AC3) · 25 Jun 2020

Thank you for the question. The overall relative errors in the retrieved Raman backscatter coefficients and linear depolarization ratios for the PollyXT lidar correspond to 5-10% and for the extinction coefficients the range is 10-20%. These uncertainties propagate to the retrieved Ångström exponents and LRs. A better description has been added due to the comment of one of the reviewers in which we have updated the text in the manuscript. In Sect 2.3 we have added: "For the mean optical properties, only regions where the extensive aerosol properties (backscatter and extinction coefficients) were nearly constant were considered. This means that within the defined layer, the variability of the optical properties should be less than the statistical uncertainty of the individual data points". Regarding the retrieval of the optical properties,

we have used the validated, under the EARLINET network, algorithm which has been developed by TROPOS team in Leipzig, Germany. Systematic errors in the lidar arise from the non-linearity of a photodetectors, errors in the calibration of the optical data or due to imperfect optical components that are sensitive to polarized light. The first two are considered through subtraction of the background signal and the correction of the dead time. Furthermore, an adequate reference height and the operation of the photo-multipliers in the linear region reduces further the two corresponding errors (Baars et al., 2016; Engelmann et al., 2016). The third systematic error is corrected through the methodology proposed by Mattis et al., 2009. As mentioned, all these systematic errors are corrected thus their contribution to the reported uncertainty is assumed to be 0. Therefore, the uncertainty in the optical parameters is solely the result of the statistical errors. In general, we try to minimize (detect and correct) the systematic errors through the mandatory annual quality assurance routines developed within EARLINET (telecover test, Rayleigh fit, trigger delay, dark measurement test and depolarization calibration) (e.g., Freudenthaler, 2009; Pappalardo et al., 2014; Wandinger et al., 2016). For example, through the telecover test, we detect misalignments in the near range while the depolarization calibration is crucial for the correct calculation of the relative amplification factor between the cross and parallel polarized channels. Thus, Fig. 7 concerns the statistical errors and the standard errors from the multiple layer statistics in each height range.

---

## Author Response (AR2)

Response to Editor:

**Regarding the Comment 4 of RC1:**

As already discussed in the response to Editor in the public discussion, the systematic errors are included/corrected in the lidar retrievals therefore their contribution to the reported uncertainty is 0. We want to thank the Editor for his persistence as it gave us the opportunity to form a comprehensive answer to this matter.

**Some of the new references added in the new text are not in the reference list:**

We have now included all the references which have been left out. Specifically, the ones missing were: Giannakaki et al.,

2020, Papagiannopoulos et al., 2016, Vaughan et al., 2009 and Kim et al., 2018. Thank you for your remark!

**Title: Have not changed according to the reviewer suggestion, as mentioned in the text:**

The title of the paper has been changed from: "Characterization of background aerosol particle properties over the United Arab

Emirates" in the initial submission to "Optical and geometrical aerosol particle properties over the United Arab Emirates."

after it. The reviewer suggested to change the title to something like "One-year lidar characterization of the vertical aerosol particle profile over the United Arab Emirates" and not include the word "background" as it was originally mentioned. The new title does not include the specific word and to our preference it describes better the content of the paper as opposed to the reviewer's suggestion.

[revised manuscript text omitted]